# A high-frequency, long-term data set of hydrology and sediment yield: The alpine badland catchments of Draix-Bléone Observatory

Sebastien Klotz[1], Caroline Le Bouteiller[1], Nicolle Mathys[1], Firmin Fontaine[1], Xavier Ravanat[1†], Jean-Emmanuel Olivier[1†], Frédéric Liébault[1], Hugo Jantzi[1*], Patrick Coulmeau[1†], Didier Richard[1], Jean-Pierre Cambon[1†], Maurice Meunier[1]

[1]Univ. Grenoble Alpes, INRAE, CNRS, IRD, Grenoble INP, IGE, 38000 Grenoble, France

*Correspondence to*: Caroline Le Bouteiller (caroline.le-bouteiller@inrae.fr)

*Now at: HYDRETUDES Grand Sud Pyrénées, 31 100 Toulouse, France

**Abstract.** Draix-Bléone critical zone observatory was created in 1983 to study erosion processes in a mountainous badland region of the French Southern Alps. Six catchments of varying size (0.001 to 22 km$^2$) and vegetation cover are equipped to measure water and sediment fluxes, both as bedload and suspended load. This paper presents the core dataset of the observatory, including rainfall and meteorology, high-frequency discharge and suspended sediment concentration and event-scale bedload volumes. The longest records span almost 40 years. Measurement and data processing methods are presented, as well as data quality assessment procedures and examples of results. All the data presented in this paper is available on the open repository https://doi.org/10.17180/obs.draix (Draix-Bleone Observatory, 2015) and a 5-year snapshot is available for review at https://doi.org/10.57745/BEYQFQ (Klotz et al., 2023).

## 1 Introduction

Mountain hydrology and sediment yield from small upland catchments have strong impacts on natural hazards, water quality and dam management. From a hazard point of view, mountain floods are often characterized by their rapid hydrological response, the presence of sediment transported both as bedload and suspended load, dramatic morphological changes, which make them difficult to anticipate and more damageable (Stoffel et al., 2016). From the point of view of water quality and biodiversity, an excess in fine sediment supply can affect river habitats by clogging the gravel porosity, reducing oxygen and light availability (Geist et al., 2007). Finally, while mountain hydrology has long been a resource for hydropower, the associated sediment fluxes and their accumulation behind dams are on the contrary a permanent issue for dam managers, that reduces dam capacity, safety and cost-effectiveness (Palmieri et al., 2001).

For all these reasons, there is a considerable interest for understanding and predicting hydrology and sediment yield from headwater catchments. However, predicting water and sediment fluxes in such environments is still difficult, due to many factors such as the spatial variability of rain in mountainous areas, the rapidity and intensity of the hydrological response

(flashfloods), the diversity and intensity of sediment transport processes (debris flows, landslides, bedload and suspended transport of varying concentrations, rapid morphological changes…) (Vanmaercke et al., 2012, Stoffel et al., 2016). Moreover, measuring water and sediment fluxes in a mountain environment can be challenging, with limited access, short response times, and high sediment yield that may damage instruments, so little data is available for process understanding and model calibration.


Because of the stochastic nature of rainfall and resulting runoff and sediment transport, long-term and continuous monitoring is required to capture the variability of water and sediment fluxes. Long-term monitoring is also necessary to track changes in the hydrological and sediment yield response in the current context of climate and land use change. However, existing long-term and continuous data sets on sediment yield from headwater catchments, including both suspended load and bedload, are

extremely rare due to all the difficulties mentioned above (Turowski et al., 2010). Suspended load only has been measured continuously in the Vallcebre catchments in the Spanish Pyrenees since 1990 (Gallart et al., 2013b), bedload only has been measured regularly at the Erlenbach catchment since 1986 (Rickenmann, 2020), and long-term combined records of suspended and bedload are available in the Rio Cordon since 1986 (Rainato et al., 2017), in the Eshtemoa catchment since 1991 (Alexandrov et al., 2009), and in the Draix catchments, which are presented here, since 1984.


Badland areas are specific landforms characterized by their highly dissected topography, scarce vegetation cover, fast hydrological response, high erosion rates and associated sediment supply (Canton et al., 2018, Moreno-de-las-Heras and Gallart, 2018). Despite a high variability among sites, badlands and especially humid badlands are among the most productive sites in the world in terms of sediment supply (Nadal-Romero et al., 2011, Gallart et al., 2013a). Consequently, even small

badland areas within large catchments may contribute significantly to the total sediment yield of these catchments. This has motivated several field studies and monitoring setups focusing on erosion and sediment transport processes in badland areas (Howard and Kerby, 1983, Canton et al., 2001, Mathys et al., 2003, Regues and Gallart, 2004, Higuchi et al., 2013, Liébault et al., 2016).

Draix-Bléone observatory was created in 1983 to study runoff, erosion and sediment transport processes that are involved in the hydrosedimentary response of small mountainous badland catchments. It is part of the French network of critical zone observatories OZCAR (Gaillardet et al., 2018) and European eLTER network. Several catchments have been equipped for high-frequency monitoring of rain, discharge and sediment fluxes (Mathys et al., 2003). The data collected in this observatory spans now more than 35 years. The specificity of this data set is the detailed record on sediment flux and more particularly,

the intensity of this flux. Both the suspended load and bedload components are measured. Sediment transport is particularly intense with suspended concentrations that can reach 800-900 g $L^{-1}$ (Le Bouteiller et al., 2021) and an average annual yield of 16000 t $km^{-2}$ $yr^{-1}$ for the most productive catchments.

In this paper, we present the core dataset of the observatory, which consists of rainfall and meteorological forcing, hydrological data for 6 stations and sediment flux data for suspended load and bedload at the same stations. Note that the data from the seventh station of Draix-Bleone observatory, the Galabre, is not presented here since it was already published (Legout et al., 2021). We first present the study site. Then for each type of data, we explain the monitoring and data processing procedures. A few examples of studies based on the data are then presented. Finally, the last section describes the public repository where the data is publicly available.

## 2 Presentation of the site

The paper focuses on catchments from Draix-Bléone Observatory, which are located in the upper catchment of the Bléone river, upstream of Digne-les-Bains, in the French Southern Alps (see Fig. 1).

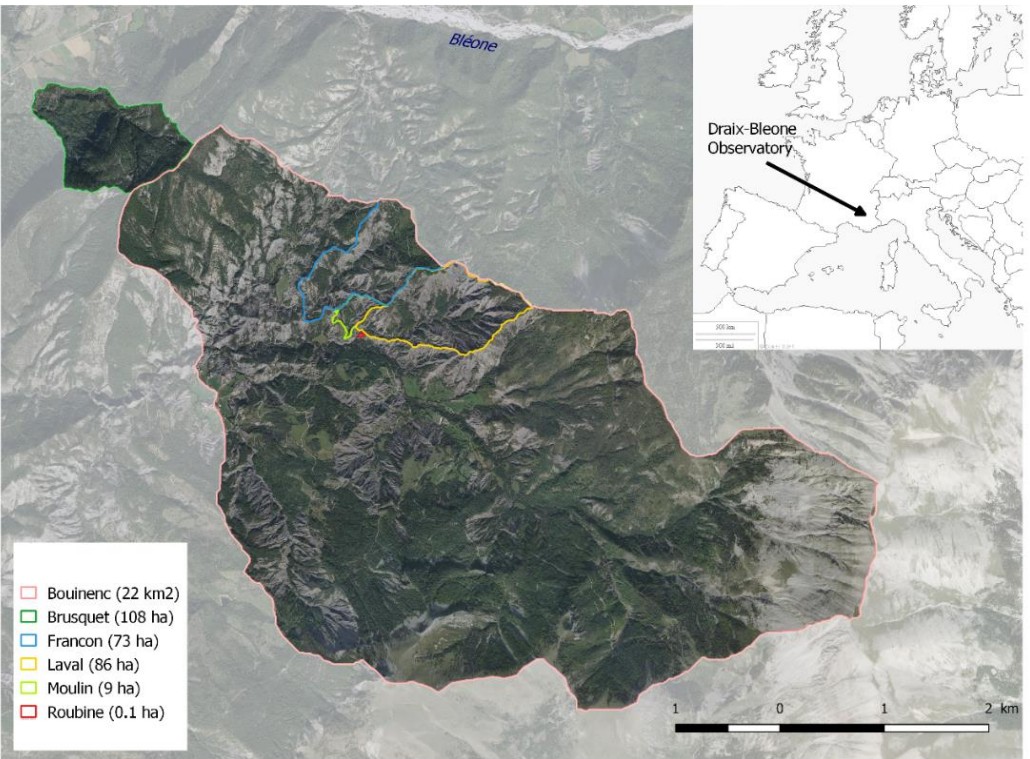

**Figure 1: Catchments of Draix-Bléone observatory (the Galabre catchment is not represented in this map). Aerial photo from BDOrtho@IGN.**

From a geological point of view, the region is characterized by a succession of limestone, marly limestone and marl layers from the Jurassic period, partially covered by Quaternary deposits. Marl outcrops locally called "Terres Noires" are places of

intense erosion characterized by badland morphology, deeply incised gullies, weathered surface and high sediment yield (see Fig. 2). At a regional scale, it has been estimated that these Terres Noires provide 40 % of the total sediment flux of the Durance river, whereas they only represent 1.5 % of its catchment (Copard et al., 2018).

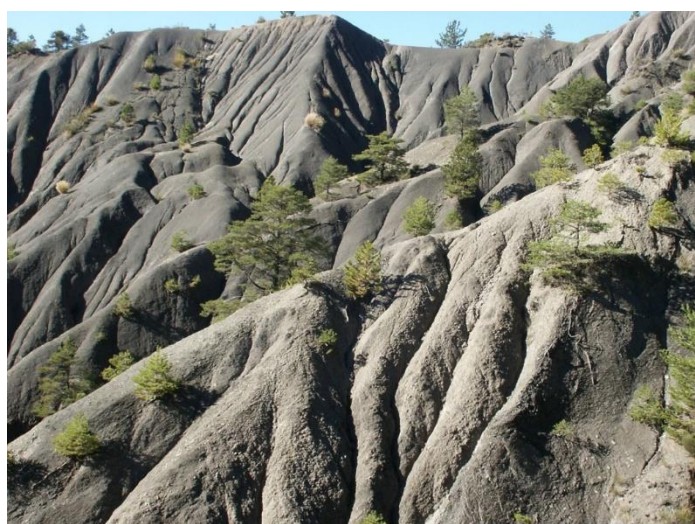

**Figure 2: Hillslopes with weathering and gullying in Draix-Bléone observatory (photo N. Mathys)**


The climate is characteristic of a mountainous and Mediterranean region with two rain periods in spring and autumn, winter frost and intense summer storms that may give rise to flash floods. Mean annual rainfall is 919 mm at the Laval raingauge over the period 1985-2022 and mean annual temperature is 9.9°C at the Plateau station over the period 2001-2022. More details on the rainfall and meteorological dataset are given in Sect. 3.1.

While most of the badland area is denuded, some hillslopes were reforested at the end of the XIX[th] century following erosion control policies (Restauration des Terrains de Montagne, RTM) in the French Alps. Natural vegetation where it is able to maintain is mostly composed of Scots pine (*Pinus sylvestris*), sea buckthorn (*Hippophae rhamnoides*), broom (*Genista cinerea*), juneberry (*Amelanchier ovalis*), and RTM reforestation was mainly based on Austrian black pine (*Pinus nigra*). Vegetation cover for each catchment is given in Table 1.

Six catchments are equipped for the monitoring of water and sediment fluxes. Their localization appears in Fig. 1 and their characteristics are summarized in Table 1. Drainage area ranges from 0.001 to 22 km². All catchments drain only marly badland areas, except for the larger catchment, the Bouinenc, that integrates a wider variability of lithologies and land cover. While badland catchments are mostly denuded, the Brusquet was reforested at the end of the XIX[th] century.

 **Table 1: Catchments names and characteristics. Average slope derived from 5-m DEM from IGN BDALTI 2016**

| Catchment | Drainage area (km$^2$) | Vegetation cover (%) | Average catchment slope (%) | Elevation (m a.s.l.) | Observed since |
|---|---|---|---|---|---|
| Roubine | 0.0013 | 21 | 48.1 | 850-885 | 1983 |
| Moulin | 0.09 | 46 | 26.7 | 850-925 | 1988 |
| Francon | 0.73 | 56 | 39.2 | 830-1140 | 1984 |
| Laval | 0.86 | 32 | 51.1 | 850-1250 | 1984 |
| Brusquet | 1.07 | 87 | 49.1 | 800-1260 | 1987 |
| Bouinenc | 22 | 75 | 47.9 | 810-2280 | 2008 |

## 3 Data

The present paper focuses on the core data set of the observatory, which includes rainfall and meteorological data, hydrological data, and sediment flux data for both bedload and suspended load. A summary of the dataset with acquisition periods is presented in Figure 3. In the following, we detail how this data is collected and processed before reaching the database. We present the type of instruments that are used, the acquisition protocol, the data filtering or gap filling procedures when applicable, the data quality assessment. Data quality is defined following a code that is defined in Table 2. These quality codes have been developed to provide a qualitative information on the reliability of the data. Quality codes 2 and 3 mean that the data is respectively of good and intermediate quality compared to standards for this type of data and can be used for quantitative analysis. Estimation of the uncertainty associated with quality 2 and 3 is given in the following sections for each type of data. Quality codes 4 and 5 mean that the data is of low quality compared to standards and can mostly be used for qualitative analysis (e.g. detection of flood events in the discharge time series). Previous quantitative analysis on Draix observatory dataset generally excluded such data. In the early years of the observatory, no quality codes were attributed, but all data of poor quality was immediately classified as missing data. Hence the remaining data with no quality code (i.e. code 1) can generally be considered to have a quality 2 or 3.

**Table 2: Quality codes of Draix-Bléone data set**

| Quality code | 0 | 1 | 2 | 3 | 4 | 5 |
|---|---|---|---|---|---|---|
| Meaning | Missing data | No quality attributed | Good quality | Intermediate quality | Uncertain quality | Poor quality |

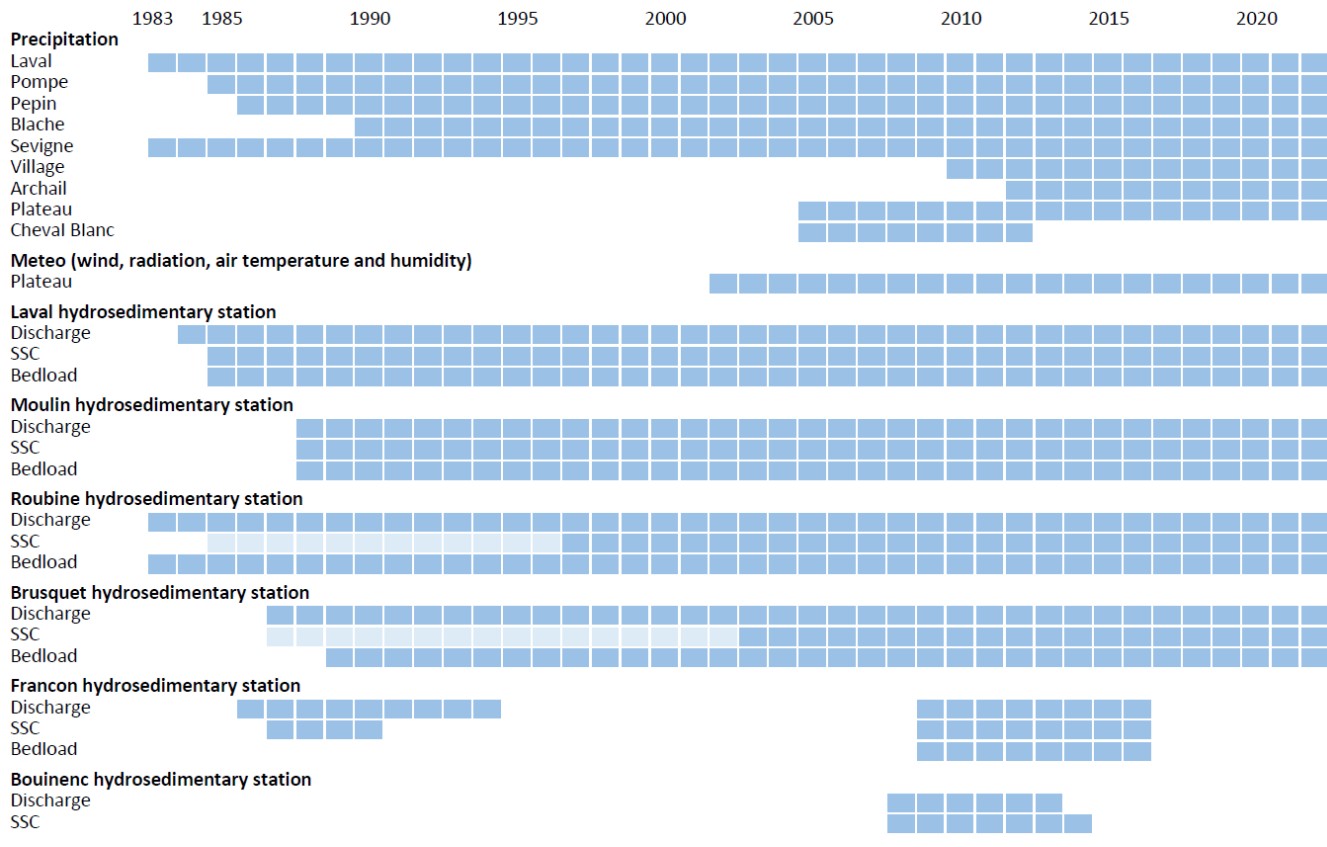

**Figure 3: Acquisition period for each station and variable. Light color indicates data that is not in the database yet.**

## 3.1 Atmosphere

### 3.1.1 Rainfall

*Measurement*

Rainfall has been measured at 9 locations, currently 8, indicated in Fig. 4, using tipping bucket rainfall recorders that allows recording each 0.1 or 0.2 mm of rain (cone surface of 1000 or 2000 cm$^2$). Table 3 lists all the raingauges, the type of instrument and logger, the coordinates and elevations, and the period of data acquisition. All the raingauges except three are equipped with an insulated chamber, and a heating system was installed only for the Laval raingauge in 2007. Data is stored in a logger system that records tipping times. All the raingauges except two are also equipped with a container that collects water under the raingauge. Every two weeks, data is retrieved from the logger, the raingauge is cleaned if needed (leaves, needles) and the volume stored in the container is measured to check that there is no bias or drift in the instrument.

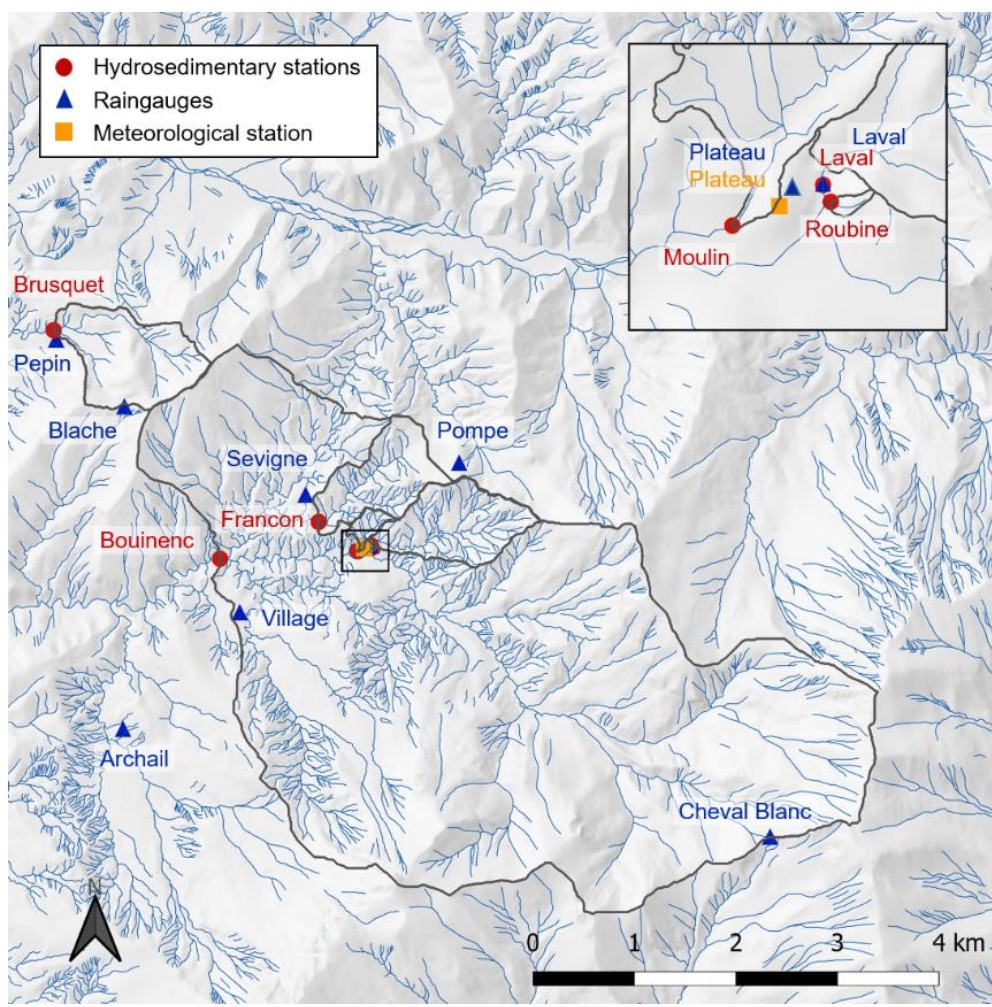


**Figure 4: Location of permanent instruments. Grey lines at the catchments delineation. The inset is a zoom on the Laval, Moulin and Roubine outlets. Hydrographic network is from BDTopo@IGN and shaded elevation is from BDAlti@IGN**

**Table 3: List of raingauges and their characteristics. Types of measuring device and data logger may have changed over time; only 135  the most recent system is listed.**

| Name | X coord (Lamb 93) | Y coord (Lamb 93) | Elevation (m.a.s.l) | Measuring period | Measuring bucket | Data logger | Resolution (mm) |
|------|---------|---------|----------|----------|----------|--------|-----------|
| Laval | 968817 | 6343670 | 849 | 1983 - now | Précis Mécanique | Alcyr Danae | 0.2 |
| Pompe | 969688 | 6344483 | 1070 | 1985 - now | Précis Mécanique | Alcyr Danae | 0.1 |

| | | | | | | | |
|---|---|---|---|---|---|---|---|
| Pepin | 965719 | 6345699 | 812 | 1986 - now | Précis Mécanique | Alcyr Danae | 0.1 |
| Blache | 966391 | 6345038 | 1115 | 1990 - now | Précis Mécanique | Alcyr Danae | 0.1 |
| Sevigné | 968176 | 6344170 | 873 | 1983 - now | Précis Mécanique | Alcyr Danae | 0.1 |
| Village | 967527 | 6343010 | 862 | 2009 - now | Campbell | Alcyr Danae | 0.2 |
| Archail | 966378 | 6341860 | 907 | 2012 - now | Précis Mécanique | Alcyr Danae | 0.2 |
| Plateau | 968755 | 6343640 | 862 | 2001 - now | Campbell | CR10X | 0.2 |
| Cheval Blanc | 972751 | 6340798 | 1730 | 2002 - 2009 | Précis Mécanique | Alcyr Danae | 0.2 |

*Data processing*

The raw data series is processed as follows: First, when the data is retrieved from the logger, we check for consistency between cumulated recorded rainfall and water volume stored in the contained below the raingauge. If these two values differ by more

than 2 % and less than 10 %, the raingauge is recalibrated, and the data for the antecedent rainfall period is corrected such that the cumulated rain over the period corresponds to the real water volume that was collected. Secondly, data from all the rain gauges are compared. Due to the small spatial extent of the site, most rainfall events reach all the raingauges. Snow events are detectable in the raw data set by the difference in the cumulated rainfall recorded by heated and non-heated raingauges. Cumulated rainfall recorded by non-heated raingauges increases slowly over the days that follow the precipitation event as

snow melts, especially during daytime, whereas cumulated rainfall recorded by the heated raingauge increases more quickly over the duration of the precipitation event. Moreover the site is also equipped with a disdrometer (data not presented here) which measures drop size and velocity distribution and allows distinguishing snow from rain events. When a snow event is detected, the data from non-heated raingauges is kept as it is with a lower quality code.

*Quality assessment*

Following Table 2, quality code "0" is attributed to all missing data. Quality code "2" is attributed to correct data. When there is a misfit smaller than 10% between the recorded rainfall and the container, the data that is stored in the database is the corrected data with a quality code "2". Quality code "3" is attributed to snow melt data over a short period, and quality code "4" is attributed to snow melt data when the snow and melting last several days. Quality codes "4" or "5" are attributed when the bucket is found to be partially obstructed by leaves or needles or for cumulated records that differ by more than 10 % from

the container. Quality code "1" generally corresponds to the oldest data that has not been qualified. A quantitative uncertainty

estimation for rainfall data with quality codes 2 and 3 is therefore respectively $\pm 2\%$ and $\pm 10\%$. Note that the rainfall data from raingauges Laval and Pompe is very similar therefore we recommend using Pompe data when Laval data is missing.

*Examples of results and treatment*

Detailed records of rainfall allow computing rainfall intensity at a short time-scale. Figure 5 shows an example of a flood from June 3rd, 2013. Rainfall intensity computed over one minute reaches 84 mm h$^{-1}$ at the Pompe raingauge and 126 mm h$^{-1}$ at the Pépin raingauge.

Rainfall records are also cumulated and analyzed at the daily, monthly and yearly scale. Figure 6 shows the monthly repartition of the rainfall over the year for the Laval raingauge. For a total annual rainfall of 920 mm, the rainiest periods are spring and autumn.


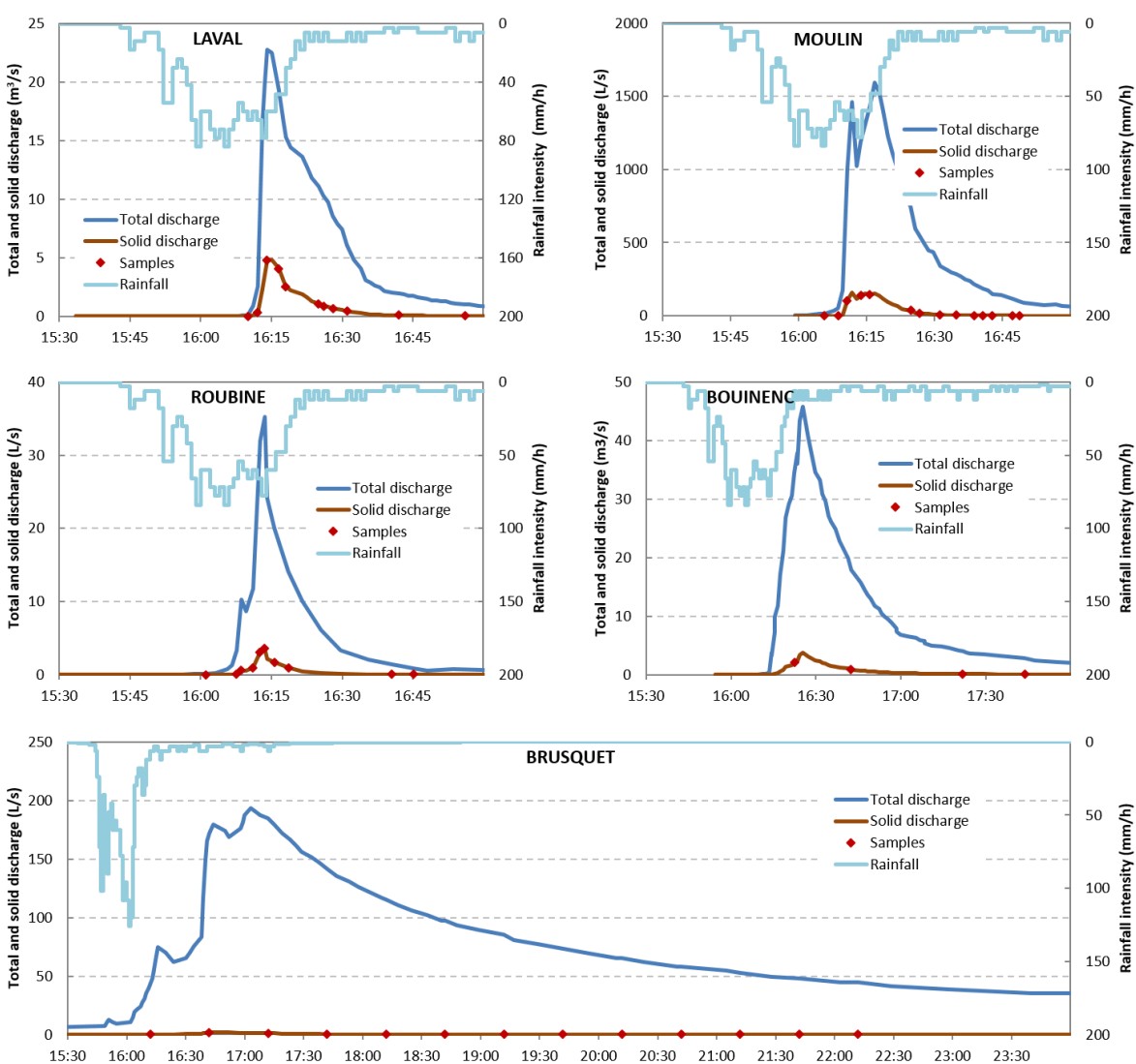

**Figure 5: Example of data for the flood of June 3rd, 2013. Rainfall intensity computed over 1-minute from the Pompe and Pépin raingauges. Water and sediment discharges from each station.**

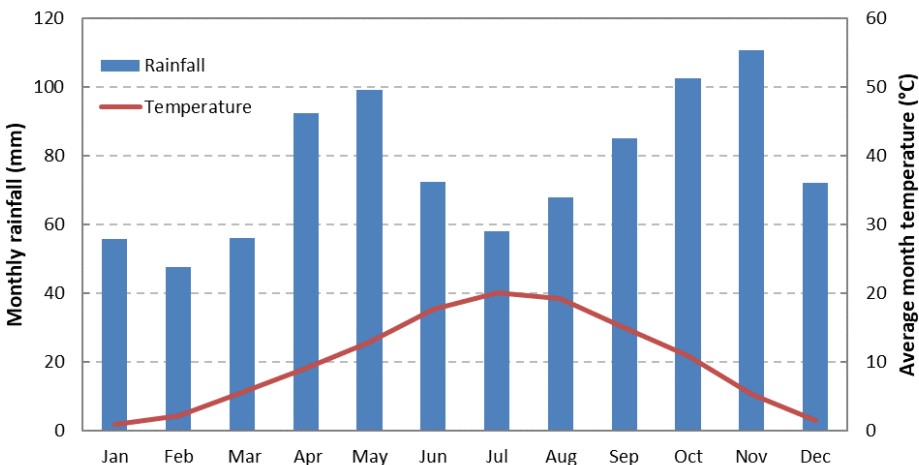


**Figure 6: Monthly rainfall at the Laval raingauge and average monthly temperature at the meteorological station. Rainfall data is averaged over period 1985-2018 and temperature data is averaged over 2001-2018.**

### 3.1.2 Meteorology

*Measurements*

A full climatic station was installed in 2000 on the ridge between the Laval and Moulin stations (L93 coordinates are 968755 and 6343638, elevation is 862 m, see location in Fig. 4). It includes temperature, air humidity, wind direction and speed, and radiation (short wave-length). The station is equipped with a Campbell Scientific CR10X data logger, solar panels and batteries.

Table 4 summarizes the type of instruments, measurements and associated acquisition periods. A second climatic station was installed in 2013 in the Brusquet watershed but the data is not validated yet therefore it is not presented here.

*Data processing*

Meteorological data is checked for consistency but does not generally require processing before storage in the database, unless if failures are detected on the instruments or logger. In such cases the data is indicated as "missing data" until the device is

repaired. The uncertainty associated with this data is therefore the measurement precision given in Table 4

**Table 4: Meteorological measurements and instruments**

| Measurement | Instrument | Precision | Acquisition period (min) |
|---|---|---|---|
| Temperature | Campbell  PT100 | ±0.1°C | 30 (mean value) |
| Humidity | Campbell  HMP45C | ±3% | 30 (mean value) |

| Radiation | Campbell SP-LITE pyranometer | | 30 (mean value) |
|---|---|---|---|
| Max wind speed (over recording period) | Campbell wind monitor 05103 | ±1% | 30 |
| Mean wind speed | " | ±1% | 10 (mean value) |
| Mean wind direction | " | | 10 (mean value) |

## 3.2 Runoff

*Measurement*

Hydrological stations are installed at the outlet of each catchment, and their location is visible in Figure 4. All the stations have the same design with a data logger (Campbell Scientific or Serosi) that controls sensor acquisition and stores the data. Some stations are connected to power supply while others are equipped with solar panels and batteries. Except for the Bouinenc, the gauging system is either a Parshall flume or a V-notch weir, and it is associated with one or several devices for water level

measurement. A summary of the gauging devices and water-level instruments for each station is presented in Table 5. Photos of all the gauging stations are presented in Figure 7.

For the Laval station, the gauging system is a Parshall flume (size 2 ft). The Parshall standard relation between water depth and discharge is used until the flume is flooded, which corresponds to a discharge of 1.9 $m^3$ $s^{-1}$. Above this value, the flow is constrained inside a rectangular weir and a weir equation is used to relate water depth and discharge (all gauging relations are

detailed in Table 6). The water level is currently measured with an ultrasonic sensor (Paratronic US6) and two "Nilomètres" (Serosi, vertical arrays of electrodes that are progressively connected as water level rises) for low and high water stages (sizes 960 and 1920 mm). Note that in the past, different devices were used to measure the water level. A bubbler sensor was mainly used before 1997, and since then, several sensors including US, radar, a home-made sensor called "ELLAN" (Olivier and Pebay-Peroula, 1995) and Nilomètre have been used. Water level measurements are recorded with a Serosi data logger with a

frequency that is controlled by water level changes. Water level is measured every 10 seconds in summer and every minute in winter, and the data is stored only when it differs from the previous record by more than a given threshold (5 mm). The resulting record has therefore a varying timestep depending on the flood dynamics. For summer floods generated by storms, a time period of 10 seconds was chosen to best capture the flood dynamics, whereas for autumn and winter slower and longer events, a time period of one minute was chosen to save some storage capacity. At every field visit (every 2 to 4 days for the Laval,

Moulin, Roubine, and every 2 weeks for the Brusquet), the discharge is manually measured with a calibrated bucket and chronometer at low flow. A standard water level scale is also installed in the Parshall flume and the water level is visually recorded at every field visit. Specific conditions such as ice on the sensors or in the flume are also recorded during each field visit.

For the Moulin station, two gauging systems are available: a trapezoidal flume and a Parshall flume of size 1.5 ft. The

trapezoidal flume is located a few meters upstream of the Parshall flume. The Parshall standard relation is used until the flume

is flooded, which corresponds to a discharge of 0.75 m³ s⁻¹. The depth-discharge relation of the trapezoidal flume is then used when the Parshall flume is flooded. Water level is measured with a nilomètre and a radar (Vegapulse WLS61) in the Parshall flume. Water level was recorded on a Serosi data logger using the same procedure as for the Laval until 2015. It is now recorded on a Campbell CR1000 logger, using a simpler procedure where one point is stored every minute.

For the Roubine station, the gauging system is a V-notch weir. The water level is measured with a nilomètre and recorded on a Serosi logger. It is measured every 10 seconds in summer (1 minute in autumn and winter) and the data is recorded only if the change in the water depth since the last measurement exceeds a given threshold (same procedure as the Laval). The discharge is computed from the water depth using the weir equation listed in Table 6.

For the Francon station, the gauging system is a Parshall flume and the water level is measured with an ultrasonic sensor
(Paratronic US10) and recorded on a Serosi logger following the same type of procedure as what was described for the Laval. For the Brusquet station, the gauging system is a Parshall flume, and the water level is measured with a nilometer (1920 mm) located in a well. The acquisition frequency is one point per minute. A V-notch weir with a nilomètre was installed in 2010 to better measure low flows but it was damaged by a flood in 2018.

The Bouinenc station is different from the others since it is located in a natural river section next to a bridge in the village of
Draix. Until 2016, the water level was measured with an ultrasonic sensor (Paratronic US10) attached to the bridge. A gauging curve was built to compute the discharge based on manual measurements of flow velocity using the salt dilution method. In June 2013, the section was greatly modified by a series of floods that moved bars around and a new bar appeared just below the US sensor. The gauging relation is therefore out-of-date. As a result, no discharge data is available after this date.

**Table 5: Gauging stations and water level measurements (only the devices that are currently in use are indicated but other have been used in the past)**

| Station name | X coord (L93) | Y coord (L93) | Elevation (masl) | Gauging system | Water level sensor (model and producer) | Acquisition frequency | Monitoring period |
|---|---|---|---|---|---|---|---|
| Laval | 968818 | 6343668 | 850 | -Parshall flume (2 ft) <br> -Rectangular weir | -Nilometer 960mm and 1920mm (Serosi) <br> -US6 (Paratronic) | Up to 10 seconds | 1984 - now |
| Moulin | 968688 | 6343610 | 847 | -Parshall flume (1.5 ft) <br> -Trapezoidal flume | -Nilometer 1200mm <br> -Radar Vegapulse WLS61 | Up to 10 seconds | 1989 - now |

| Roubine | 968828 | 6343644 | 852 | V-notch weir | -Nilometer 1200mm | Up to 10 seconds | 1983 - now |
|---------|--------|---------|-----|--------------|-------------------|------------------|------------|
| Francon | 968306 | 6343896 | 851 | Parshall flume (2 ft) | -US10 Paratronic | Up to 10 seconds | 1986 - 1994 and 2009 - 2016 |
| Brusquet | 965694 | 6345789 | 801 | -Parshall flume (2 ft)<br>-V-notch weir | -Nilometer 1920mm | Up to 1 minute | 1987 - now |
| Bouinenc | 967333 | 6343536 | 798 | Natural, gauging curve | -US10 Paratronic | Up to 1 minute | 2008 - 2013 |

**Table 6: Gauging relations for each station**

| Station name | Gauging equation with H water depth (m) and Q discharge (L/s) |
|--------------|-------------------------------------------------------------|
| Laval | $For\ H < 1.2\ :\ Q = 1428 * H^{1.5493}$<br>$For\ H > 1.2\ :\ Q = 1900 + 14680 * (H - 1.2)^{1.5}$ |
| Moulin – Parshall flume<br>Moulin – trapezoidal flume | $For\ H < 0.8\ :\ Q = 1.057 * H^{1.5381}$<br>$Q = 0.2089 * H^{1.4409}$ |
| Roubine | $For\ H < 0.23\ :\ Q = 0$<br>$For\ H > 0.23\ :\ Q = 1420 * (H - 0.23)^{2.5}$ |
| Brusquet and Francon | $For\ H < 1.2\ :\ Q = 1428 * H^{1.5493}$<br>$For\ H > 1.2\ :\ $ Discharge not estimated |
| Bouinenc | $Q = 12445 * (H)^{1.7164}$ |

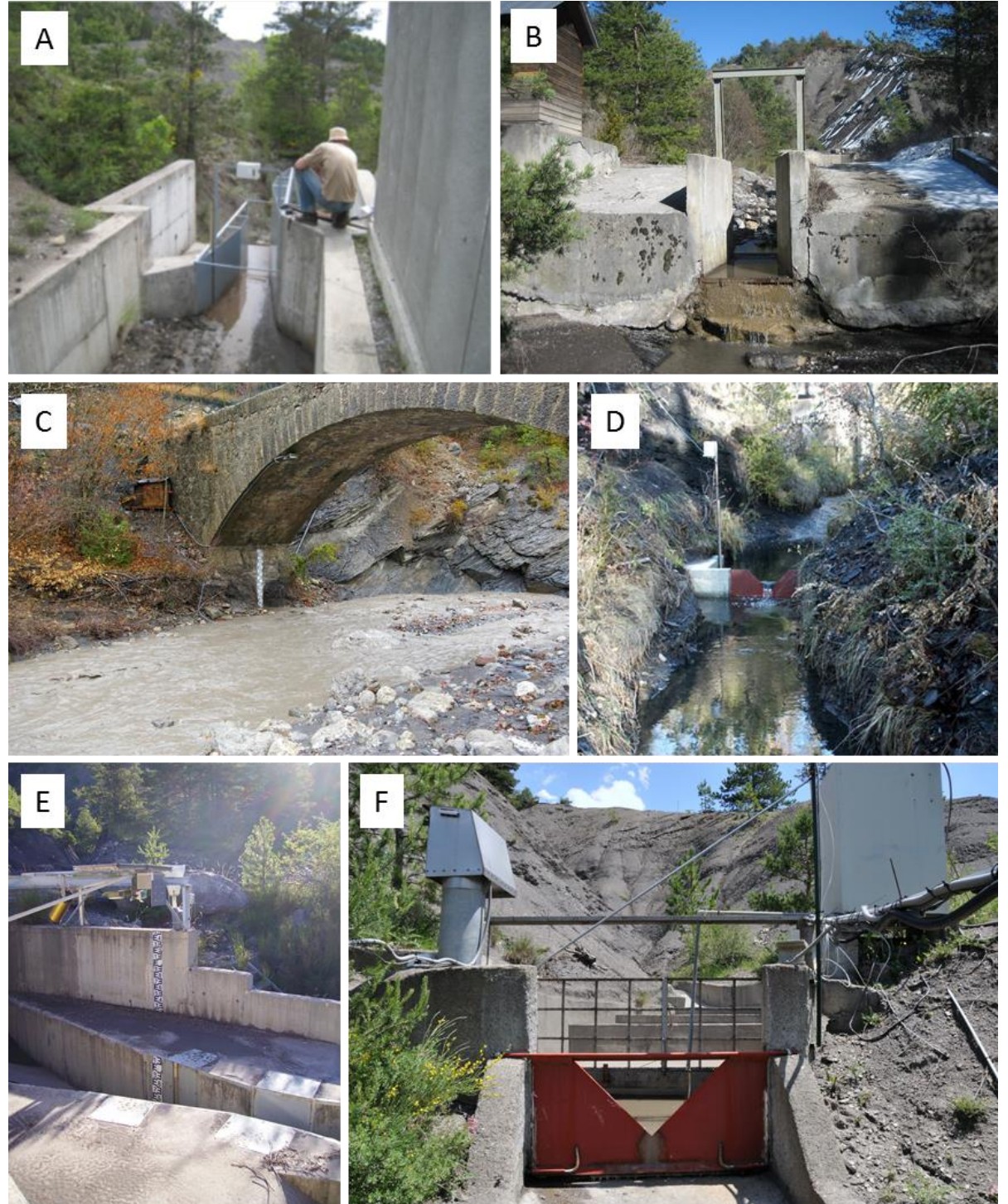


**Figure 7: Photos of all the stations: A) Moulin Parshall flume. B) Francon Parshall flume looking upstream. C) Bouinenc natural gauging section. D) Brusquet low-discharge weir. E) Laval Parshall flume. F) Roubine weir looking upstream (photo F from J. Latron)**

*Data processing*

When available, the nilomètre is the reference sensor for the water depth. It is completed or corrected by data from other sensors only when needed. The data processing scheme for the water depth record is the following:

- Remove human artefacts from the record. In particular, cleaning operations in the sediment deposit area upstream of the gauging station and on the grid that dams this deposit area generally generate short peaks of water depth in the data record, which does not correspond to actual flows. These points are therefore removed from the record.

- Apply a 5-point running mean filter to remove noise due to surface ripples and raindrop impacts on the water surface.

- Finally, add scale readings and manual bucket measurements to the record. In case of inconsistency, the scale readings and manual bucket measurements are considered more reliable than the sensor record. If there are several scale readings or bucket measurements that indicate a similar misfit with the sensor data, the sensor data is corrected to fit the manual record (shifted), otherwise, the sensor data is kept with a lower quality code.

- The water depth is then converted in discharge using the gauging relation of the station.

*Quality assessment*

Regular field visits are a great advantage for data quality, since they allow manual measurements of low discharges and scale reading, and field observations, that are useful to interpret, criticize and possibly correct the sensor data. Several issues may affect the quality of the water depth record. When sediment or ice are deposited next to the measurement area, when the

gauging system is flooded, or when the deposit trap is full of sediment, a lower quality is associated with the water depth data (quality code 3). This is also the case when the sensor record is very noisy or differs significantly from the scale reading. When none of these issues have been noted, the data is considered of good quality (code 2) and the uncertainty is mostly related to the noise from ripples and raindrops. Old data was not given any quality code. The uncertainty in the gauging relation is also accounted for in the quality code attributed to the discharge data. In particular, at the Laval station, when the Parshall flume is

flooded, a rectangular weir gauging relation calibrated only from numerical modelling is used to estimate discharge (Table 6). For this reason, a quality code 3 is attributed to all discharge data above 1.89 m$^3$ s$^{-1}$. Current work based on flow surface image analysis is ongoing to improve the accuracy of high-discharge estimates. Another source of possible uncertainty for the highest flood peaks could be a buffer effect due to the sediment trap located upstream of the station. Based on our current knowledge, quantitative estimates of the discharge uncertainty are therefore 10% and 30% respectively for data with quality code 2 and 3.

*Example of results*

An example of hydrographs obtained for all the catchments on June 3$^{rd}$, 2013 is given in Figure 5. The response time for this summer event is small, ranging from 6 to 10 minutes for the mostly denuded badland catchments (Roubine, Moulin and Laval), slightly higher for the larger Bouinenc catchment (15 minutes) and much larger for the reforested Brusquet catchment (1 hour). Peak discharge reaches 35 m$^3$ s$^{-1}$ km$^{-2}$ for the smallest Roubine catchment.

### 3.3 Sediment

#### 3.3.1 Suspended load

*Measurement*

Suspended sediment concentration (SSC) is measured at all the hydrosedimentary stations using a combination of turbidity sensors and automatic samplers (see Table 7). All the stations are equipped with an ISCO automatic sampler, even two for the Laval. Each sampler is able to collect up to 24 one-liter samples. The samplers are controlled by the data logger to collect water (including suspended matter) when water depth exceed a threshold (e.g. 20 cm at Laval), remains higher than this threshold for more than half an hour, or when water depth changes faster than a threshold (e.g. 10 cm per minute at Laval). The samplers are able to collect up to one sample every minute. The bottles are retrieved after each flood and brought to the laboratory for decantation, oven-drying (24 hours at 105°C) and weighing. SSC is then calculated as the ratio of the dry weight to the sample volume. As an example for the Laval station, more than 1000 samples were collected over the period 2017-2019, representing an average of 11 samples per flood.

Moreover, the stations are equipped with sensors for a continuous measurement of turbidity. The Laval, Moulin and Francon (until 2018 for Francon) stations are equipped with an optical fiber sensor that was specifically designed for the high concentrations that are encountered in these catchments (several hundreds of g $L^{-1}$; Bergougnoux et al., 1998). This sensor is able to measure concentrations in the range 10-800 g $L^{-1}$ with an accuracy of $\pm$ 5 g $L^{-1}$. The Bouinenc and the Roubine are equipped with WTW turbidimeters (VISolid 700IQ). For the Laval, Moulin and Francon, the measurement from the optical fiber sensor is recorded only if it differs of more than a given threshold from the previous one, with an algorithm similar to what is used for water depth recording. The acquisition frequency for the turbidimeter is one point per minute at the Roubine and one point every 10 minutes at the Bouinenc station. For low flows, the water depth is not sufficient to reach the turbidimeter or the suction strainer of the sampler pipe. The concentration measurement is therefore only available for flood events.

Following the 2013 flood at the Bouinenc station, the turbidimeter and the suction pipe for the sampler have been buried in a gravel bar therefore the concentration is not measured anymore at this station.

*Data processing*

Common practice for concentration measurements consists in using the discrete concentration values obtained from the sampler to calibrate a turbidity/concentration relation (or voltage/concentration), then convert the full "continuous" turbidity time series in a concentration time series based on this relation. In Draix however, the relation between turbidity (from the WTW turbidimeter) or voltage (from the optical fiber sensor) on one hand, and concentration on the other hand, is often noisy and varies a lot from one flood to the other. In particular, a hysteresis is frequently observed between the water discharge and the concentration on one hand, and between the tension and the concentration on the other hand. Moreover, the optical fiber sensor is not reliable at low concentrations (i.e. less than 10 g $L^{-1}$). On the contrary, the concentration measured from the samples is a reliable source of information for which the uncertainty remains small especially for high values of concentration (1 to 5 % for high and low concentrations respectively). Therefore, we have defined a specific procedure to reconstruct the

concentration time series based on the discrete concentration values measured in the samples. This procedure is meant to give more credit to sample concentration data than turbidimeter data and ensure that the reconstructed concentration is equal to the sample concentration where samples are available, while preserving the hysteretic pattern between discharge and concentration. For each flood, the concentration time series is reconstructed, starting from the sample values, then combining several procedures, as illustrated in Figure 8:

- First, an event-specific concentration/voltage relation is built for the event if at least 4 samples are available. If not, the yearly or interannual relation is used.

- Secondly, the samples are represented in a discharge/concentration diagram, which highlights the hysteresis relationship of the flood (orange points in Figure 8a).

- Third, a few turbidimeter concentration data points (obtained from the concentration/voltage relation) are added to the diagram to capture the flood dynamics, i.e. extreme values, discharge peaks, inflexion points, where samples are not available (green triangles in Figure 8a). This procedure can be used only when the water depth is sufficient to submerge the optical sensor, and when the concentration is higher than 10 g $L^{-1}$.

- Fourth, the concentration is linearly interpolated following the discharge between all existing points in the diagram. This interpolation is possible only for short periods where the discharge/concentration relation is monotonous, but is not reliable over a discharge peak (plain grey lines in Figure 8a).

- Finally, a linear interpolation following the time is used for the end of the flood recession, after the last sample and when the concentrations are not measurable anymore by the turbidimeter (dotted grey line in Figure 8a).

Note that in the early years of the observatory, the suspended load was only measured from samples. The concentration time series was therefore reconstructed using a linear interpolation between the sampling points in a discharge/concentration diagram. On the other hand, when no samples are available for a flood, we simply use the annual calibration curve to convert turbidity into SSC.

Finally, a flux of suspended matter is computed by multiplying the discharge and the concentration, and a total mass exported for each flood is computed by integrating this flux over the duration of the flood. This mass can be transformed into a volume of eroded rock using the marl density value of 2750 kg $m^{-3}$.

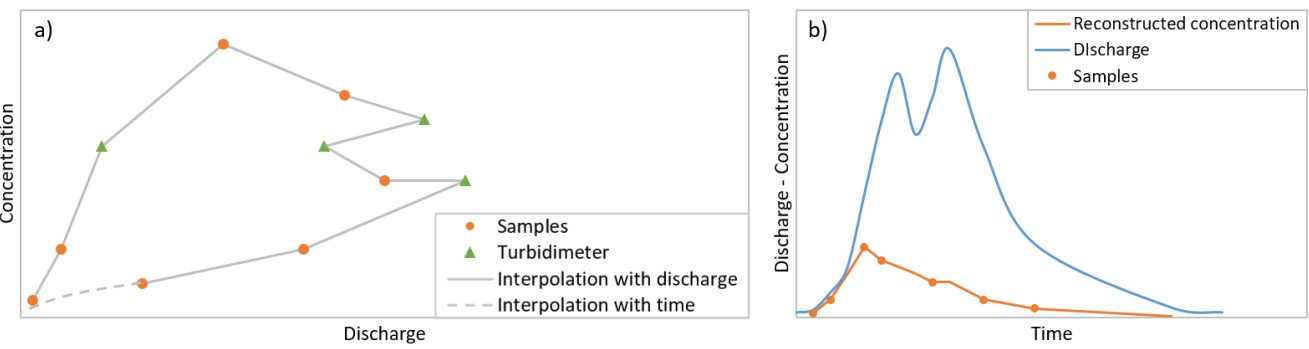

**Figure 8: (a) Reconstruction of the concentration time-series in the discharge-concentration diagram and (b) time-series of discharge and reconstructed concentration**

*Quality assessment*

A full quantitative assessment of the uncertainty associated with the SSC is currently in progress but requires detailed investigations and modelling that are out of the scope of this paper. However we provide below some elements to estimate this uncertainty and help the reader use this data. First, the measurement uncertainty associated with the concentration measured in the samples ranges from 1% to 5%. Secondly, the interannual calibration curve between turbidity and concentration measured in samples at the Laval station over the period 2012-2019 yields a mean absolute error of respectively 60% and 13% for points of concentration lower than 50g/L and higher than 50g/L. Note that at the Laval station, 80% of the annual sediment flux is exported at concentrations higher than 50g/L (Seve, 2020). As explained in Bergougnoux et al (1998), part of this error is related to the time needed to pump the water sample, which is longer than the duration of the turbidity measurement, and another part is linked to the variation in the sediment grain-size during one flood and from one flood to the other. Thirdly, a preliminary analysis over a few event-scale turbidity calibration curves suggests that the mean absolute error when estimating SSC from turbidity using these curves is approximately ±10%.

When no samples are available for a flood, the annual or interannual turbidity calibration is used and the resulting SSC time series is given quality code 3. When at least 4 samples are available and are well distributed over the flood duration and over the range of discharges, an event-scale calibration is used and combined with the sample data and the resulting SSC time series is given quality code 2. If the samples are less than 4 and/or not well distributed over the flood, a quality code 3 is given. Based on our current knowledge, a rough estimate of the uncertainty for SSC data is therefore ±10% for data with quality code 2 and ±13% to ±60% for data with quality code 3 with concentration respectively lower or higher than 50g/L.

Note that the quality of the sediment concentration data has been assessed only recently therefore most historical data in the data base has no quality code attributed. For this data, we therefore recommend considering an uncertainty of ±60% for concentrations lower than 50g/L and ±13% for concentrations higher than 50g/L, although we hope to refine and reduce these values in the future by a more detailed assessment of uncertainty which is in progress.

 **Table 7: Suspended sediment concentration and turbidity devices**

| Station name | Turbidity and SSC devices (model and producer) | Acquisition frequency | Monitoring period |
|---|---|---|---|
| Laval | - Optical fiber sensor (Bergougnoux et al, 1998)<br>- 2 ISCO automatic samplers | Up to 10 seconds | 1985 - now |
| Moulin | - Optical fiber sensor<br>- ISCO automatic sampler | Up to 10 seconds | 1988 - now |
| Roubine | - WTW turbidimeter (VISolid 700IQ)<br>- ISCO automatic sampler | Up to 10 seconds | 1985 - now |
| Francon | - Optical fiber sensor<br>- ISCO automatic sampler | Up to 10 seconds | 2010 - 2018 |
| Brusquet | - ISCO automatic sampler | Up to 1 minute | 1987 - now |
| Bouinenc | - WTW turbidimeter (VISolid 700IQ)<br>- ISCO automatic sampler | Up to 1 minute | 2008 - 2014 |

*Examples of results*

An example of suspended sediment flux time series obtained for the flood of June, 3$^{rd}$, 2013 is given in Fig. 5. Concentration reaches 594 g L$^{-1}$ at the Laval station and 30 g L$^{-1}$ at the Brusquet station. Summing up with the volume exported as bedload, the total sediment yield for this flood is 7840 tons for the Laval, corresponding to a surface ablation of 5 mm.

The highest concentrations are observed at the Laval station, and instantaneous concentrations higher than 600 g L$^{-1}$ are have been observed 11 times for the last 30-years period (see Fig. 9). For such values, the fluid becomes non-Newtonian, which might result into an overestimation of the discharge when combined to a low water depth, i.e. in the early moments of floods (Le Bouteiller et al., 2021). This effect has not been accounted for in the depth-discharge relations and might affect the discharge estimation in the earliest moments of highly concentrated floods. However, we tested this effect for the flood of July 6$^{th}$, 2006 at the Laval station, and found that the resulting error at the flood-scale was less than 1% for the total water volume and less than 3% for the total sediment volume.

Between flood events, the water depth is not sufficient to perform concentration measurements. However, for the small catchments of Laval, Moulin, Brusquet, Francon and Roubine, base flow is low or null therefore most of the discharge passes during floods and the sediment flux between the events should be negligible. This was checked for the Laval: we compared the yearly sediment export obtained by summing only the contributions of the floods, or by adding a contribution of the interflood periods assuming an average concentration of 0.1 and 1 g L$^{-1}$ (which is a high value of concentration for low flows) for these periods. An interflood contribution of concentration 0.1 g L$^{-1}$ (respectively 1 g L$^{-1}$) was found to increase the yearly sediment flux by only 0.12 % (respectively 1.2 %). It is therefore reasonable to sum the flood contributions to estimate the yearly sediment export from such catchments.

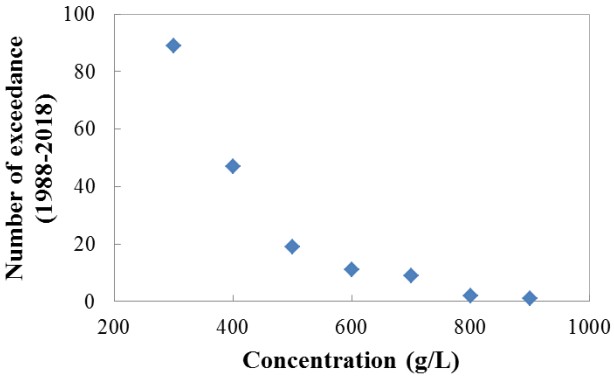

**Figure 9: Number of flood events where instantaneous concentration in suspended sediments exceeded a given value over the period 1988-2018 at the Laval station**

### 3.3.2 Bedload

*Measurement*

Sediment transport does not occur only through suspended matter but also as bedload. Apart for the Bouinenc, all the stations are equipped with a sediment trap located upstream of the gauging station. An example of the Laval sediment trap is presented in Fig. 10. Some traps are dug into the marl bedrock (Laval, Francon) while others have a concrete bottom (Moulin, Roubine). The storage capacities of these traps are summarized in Table 8. The downstream side of the trapping area is closed by a grid. Coarse sediment transported as bedload is deposited in the traps due to the reduction of the transport capacity (flow divergence and slope reduction) and/or stopped by the grid. After each flood, or after a few consecutive flood events, the topography of the deposit area is measured and compared with the previous topography to estimate the volume of sediment that has been deposited by the flood. For the Laval, Moulin, Francon and Brusquet, the topography is measured using a tacheometer pointing at orange spray dots painted at the surface of the deposit. Approximately 120 dots are used for the Laval deposit trap, with one transect every 4 meters, and 70 dots for the Moulin deposit trap, with one transect every 2 meters. For the Roubine, the topography is measured with a rule across the deposit, with a resolution of 60 measurements for 17.7 m$^2$. The traps need to be emptied regularly, a few times a year. This is done with power shovel for the largest traps and manually for the Roubine, and the total volume excavated is also recorded at that time (number of buckets or number of trucks filled with sediments). Note that some fine sediments (<20%) are stored together with coarser material in the trap, as shown by Liébault et al. (2016), but they only represent a small fraction of the total suspended sediment volume at the event scale.

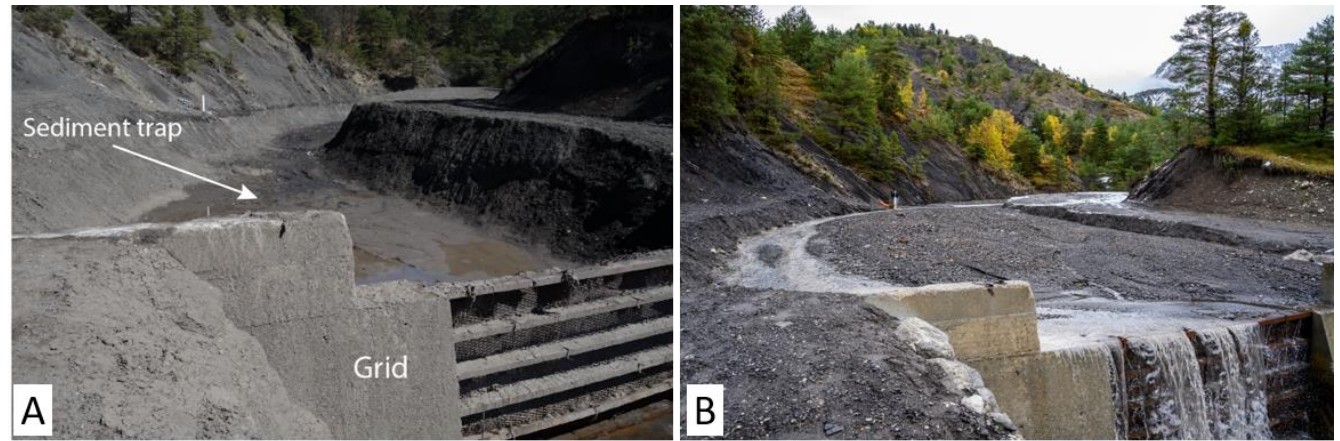

**Figure 10: Sediment trap at the Laval station. (A) Trap is empty and (B) trap is full. (photo A from C. Ariagno and B from H. Raguet)**

**Table 8: List of sediment traps and their capacity**

| Station name | Sediment trap capacity ($m^3$) | Measured since |
|---|---|---|
| Laval | 1400 | 1984 |
| Moulin | 120 | 1988 |
| Roubine | 6 | 1983 |
| Francon | 600 | 2010 |
| Brusquet | 30 | 1988 |

*Data processing*

For tacheometer measurements, the x, y and z coordinates of the painted dots are interpolated into a gridded surface with a grid size of 50 cm for the Laval, 16 cm for the Moulin. Subtraction between two consecutive surfaces over the extent of the sediment trap yields a volume of sediment which is attributed to the flood event. In the case where it is not possible to measure

the topography between one flood and the next one, the volume is attributed to the batch of floods that occurred between the two consecutive measurements of the topography. The volume is then transformed into a mass of sediments, using a bulk density of 1700 kg $m^{-3}$. This value was obtained from in-situ measurements (Mathys, 2006) on samples collected in the trap and accounts for the porosity and water content of the deposit. The trap is emptied regularly, and at every cleaning, the cumulated volume obtained by summing up the contributions of all floods since the previous cleaning is compared to the

volume that is removed from the trap.

*Quality assessment*

The uncertainty of bedload volume measurement results from the uncertainty in measuring the x, y and z coordinates of the points at the surface and interpolating these points into a surface model. Moreover, subsidence of the bedload volume is

possible as the material dries after being deposited by a flood. The uncertainty for the bedload volumes can therefore be estimated at ±10%

*Example of results*

Summing up the suspended load mass and the bedload mass for each event provides a series of event-scale total sediment yields. Figure 11 shows an example of the cumulated suspended and bedload yield for 2013 at the Laval station. The total yield for 2013 is 20 900 tons, which corresponds to a distributed ablation of 13.4 mm of marl across the catchment (computed for a fresh bedrock of density 2750 kg m$^{-3}$). Note that 2013 was one of the most erosive years recorded on the observatory. The contribution of the suspended load and bedload are respectively 72 and 28 %. The largest contribution for this year is the flood of June 3$^{rd}$ that was presented in Fig. 5. Averaging over all the period 2003-2022, the respective contributions of suspended load and bedload for the Laval station are 32 and 68%.

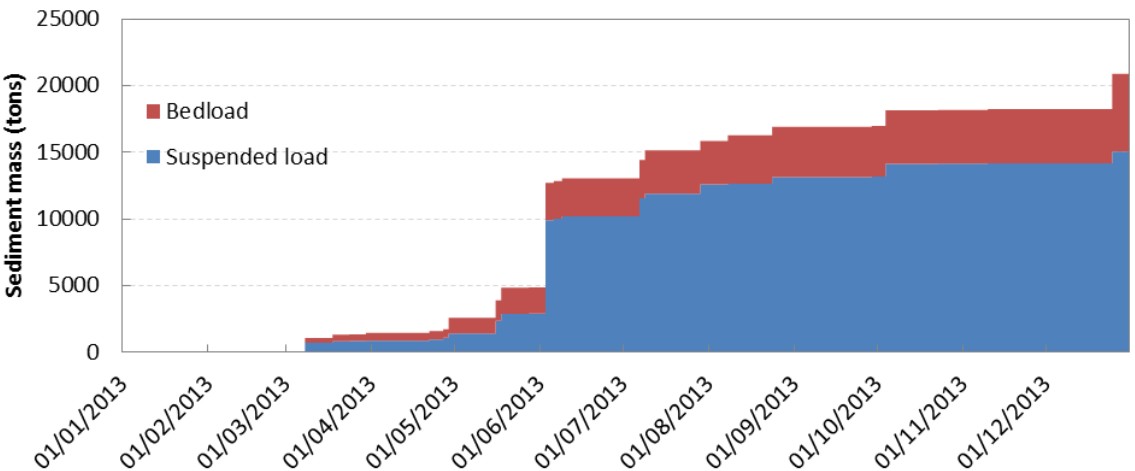

Figure 11: Cumulated yield of suspended load and bedload sediment for year 2013 at the Laval Station.

**4 Examples of studies and open questions**

The data set has been used for several studies focusing on runoff and erosion. A few examples are given below to demonstrate the potential of the data.

Detailed records of rainfall have made it possible to study the effect of rainfall intensity on sediment yield and more particularly on the initiation of bedload transport. It clearly appears that the erosive response is governed by the intensity rather by the total amount of rain, and specifically by the intensity computed over very short periods (Mathys et al., 2005, Badoux et al., 2012, Ariagno et al., 2022). This demonstrates the need for high resolution rainfall records in mountainous upland catchments.

Secondly, sediment yield from the Terres Noires is a crucial problem for managing the dams located downstream in the Durance watershed. This has motivated several modelling studies that aimed at testing the ability of models to predict sediment

yield from these catchments. Mathys et al. (2003) tested a semi-distributed model ETC and demonstrated its capacity for event-scale sediment yield prediction on the Laval catchment, provided that an appropriate hillslope supply is used. Luckey et al. (2000) used the Shetran distributed and physically-based model to simulate hydrology and sediment fluxes in the Laval, then in a virtually reforested Laval. They found that uncertainty in the parameters has a strong impact on the model predictions.

Duvert et al. (2012) found a meaningfull relation between peak discharge and event-scale sediment yield that could in turn provide moderately accurate estimates of annual sediment yield. Carrière et al. (2020) showed that a long-term landscape evolution model was able to predict reasonable annual sediment yields at the Laval but did not perform well on the Brusquet catchment because of the small and highly variable annual number of events. Ariagno et al. (2022) found an hysteretic relationship between monthly rainfall and monthly sediment export, suggesting that sediment export depends not only on the

event-scale precipitation but also on catchment-scale sediment availability, which varies over the year and can be related to winter frost-weathering. Predicting sediment yield on such catchments, and specifically at the event scale, is therefore still an open question, which requires both a better understanding of slope processes and appropriate tools to simulate the dynamics of flood events.

Thirdly, several studies have focused on the comparison between the Laval and Brusquet catchments that have similar size

and morphology but contrasted vegetation cover. From a hydrological point of view, Cosandey et al. (2005) observed a reduction of peak discharges and flood volumes at the event scale, and a slightly higher loss by evapotranspiration for the forested catchment at the annual scale. From an erosion point of view, Carrière et al. (2020) showed that the vegetation impacted strongly erosion mostly by increasing soil cohesion, and that this could explain the difference of two orders of magnitude between annual sediment yield in the Brusquet and Laval. The long record of water and sediment fluxes for these

two contrasted catchments is clearly an opportunity for a better understanding of the effect of vegetation on the water cycle and on erosion processes.

On top of the core data set described here, several other instruments and measurements have been deployed on the observatory, leading to a better understanding of the hydrological and sediment processes. Several examples of such data and measurements are listed below, even if there are not part of the present paper, because they have been made possible thanks to the detailed

knowledge and data on hydrological and sedimentary fluxes available in Draix-Bléone catchments. Rainfall simulation campaigns have underlined the role of rainfall intensity on runoff and erosion (Mathys et al., 2005). A disdrometer has been installed to capture rain drop size and velocity distributions and relate them to rainfall erosivity. Distributed soil moisture campaigns have underlined the relation between geomorphology and soil moisture dynamics (Mallet et al., 2020). Soil temperature at several depths and locations has been monitored since 2004 and related to weathering processes in the marl

(Ariagno et al., 2022). High-resolution repeated Lidar surveys have demonstrated the seasonality of slope erosion processes (Bechet et al. 2016). Detailed measurements of bedload sediment transport have been undertaken with a Birkbeck slot sampler in the Moulin ravine (Liébault et al., 2016) and analysed in relation to the seasonality of bedload transport (Liébault et al., 2022). Instantaneous bedload fluxes recorded by this sampler are amongst the highest reported values in the world, with values commonly exceeding 10 kg m$^{-1}$ s$^{-1}$. This bedload dataset also shows a counter clockwise seasonal bedload hysteresis that can

be attributed to a yearly sediment pulse reaching the catchment outlet during autumn and early winter, as also documented by scour chain surveys of alluvial deposits. Water chemistry sampling campaigns have explained the partitioning of water through runoff and subsurface flow (Cras et al. 2007), and geochemistry analyses of suspended sediment have quantified their contribution to fossil organic carbon delivery (Graz et al., 2012). Finally, in-situ $CO_2$ chambers (Soulet et al., 2018) have quantified the $CO_2$ emissions due to marl weathering and their dependency on temperature (Soulet et al., 2021). All of the

approaches listed above underline the potentiality of the dataset presented here to contribute to a variety of questions concerning critical zone processes and the interactions between chemical, physical and biotic components in these badlands. Future studies may therefore focus on exploring modelling strategies to improve sediment yield and erosion prediction, from the event-scale to longer time-scales, tracking the effects of climate change on hydrology and sediment processes in the last 40 years in order to better predict future badland response, addressing the coupling between vegetation and erosion, subsurface

water flow, physical and geochemical weathering, etc.

Data collection will continue since Draix-Bleone observatory has been labelled as SNO (National Observation Service) meaning that French institutions (CNRS and INRAE) are willing to fund it for long-term observation. Objectives for the next years include maintaining existing datasets and particularly securing bedload measurement at the Laval station, validating and making existing datasets available to the community (e.g. soil moisture and rain drop size and velocity distributions),

developing water chemistry measurements on a regular basis, testing surface image velocimetry to refine the estimation of the highest discharges, and build detailed maps of vegetation cover on the catchments.

## 5 Data availability

A snapshot of the full data set presented here is available directly for review at https://doi.org/10.57745/BEYQFQ (Klotz et al., 2023). This subset covers years 2015 to 2019 and includes rainfall and meteorological data, as well as discharge and

sediment yield data at 4 stations. The full data set with enhanced data services is available in the BDOH database repository at https://doi.org/10.17180/obs.draix (Draix-Bleone Observatory, 2015). BDOH database as well as the snapshot files contain data that has been criticized and validated following the procedures that are described above. BDOH web interface provides direct and free access to data for search and visualization. It includes detailed information on the acquisition period and number of data points, data acquisition procedures and sensors, data completeness at the monthly and annual scales and a quick

visualization tool that makes it easy for the user to browse through the data set (Branger et al., 2014). Note that data download from BDOH requires creating an account, which is free of charge but restricted to the BDOH data policy including non-commercial use of the data. The download interface offers an integrated interpolation/averaging tool that allows exporting the data either with its native resolution or with standard resolutions (for instance, hourly or daily data) useful for modelling purposes. Finally, BDOH database is also continuously updated with recent data, and is able to keep track of possible

modifications in gauging relations. A complementary spatial dataset with catchment boundaries, DEM and instrument locations is available at https://doi.org/10.57745/RUQLJL.

## 6 Conclusions

We presented a data set of hydrological and sediment fluxes from 6 small mountainous badland catchments of Draix-Bléone Observatory. Data spans up to 40 years, with a particular effort devoted to measure floods, since these short periods are

responsible for most of the sediment fluxes. High-frequency rainfall records at 9 locations make it possible to quantify short-period rainfall intensity and analyse rainfall spatial variability. Water discharge measurements provide detailed information on flood dynamics and flashiness. Both suspended load and bedload contributions are measured and indicate intense transport and erosion rates. We also presented the data criticism and validation steps that are performed in order to provide a qualified and usable data for the user.

**Author contributions**

The observatory was set up in the early 1980s by M. Meunier and J.P Cambon, and further developed by N. Mathys, JE Olivier and D. Richard, in the 1990s. JE Olivier, S. Klotz, X. Ravanat and F. Fontaine have installed and maintained the instruments and infrastructures. JE Olivier, P. Coulmeau, S. Klotz and H. Jantzi have collected the data. S. Klotz, N. Mathys and C. Le Bouteiller have criticized and validated the data with the help of F. Liebault for the Moulin data. S. Klotz and C. Le Bouteiller

wrote the paper with inputs from N. Mathys, D. Richard and F. Liebault.

**Competing interests**

The authors declare no conflict of interest.

**Acknowledgments**

Draix-Bléone observatory is supported by the Institut National de Recherche pour l'Agriculture, l'Alimentation et

l'Environnement (INRAE), the Observatoire des Sciences de l'Univers de Grenoble (OSUG), and the Institut National des Sciences de l'Univers (CNRS-INSU). It is part of OZCAR research infrastructure, which is supported by the French Ministry of Research, French Research Institutions and Universities. In the early years of the Observatory, the design, construction and maintenance of infrastructures were also supported by ONF-RTM.

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
