# Peer review of "A high-frequency, long-term data set of hydrology and sediment yield: The alpine badland catchments of Draix-Bléone Observatory"

_Earth System Science Data, 2023_

## Author Comment (AC1)

**Response to reviewer 1**

Klotz et al. have posted a rich set of meteorological, discharge and sediment yield data for 6 French alpine catchments containing badlands from the Draix-Bléone Observatory. Given the difficulty of conducting such monitoring in a mountainous environment, the temporal span of the proposed chronicles makes this dataset quite unique and therefore very valuable. The authors have extensively illustrated, via previously published studies, the potential of the proposed dataset to contribute to a variety of issues related to runoff and erosion in this specific mountainous environment.

The article is well structured and clear. The data are readily accessible and usable in their current format and size, either by using the 5-year snapshot that is available for review or by using the BDOH web interface for the full dataset (requires registration, but has powerful features that make exploration of the dataset quite user-friendly).

I very much appreciated the significant effort made by the authors to explain the complex procedures for collecting, processing and qualifying hydrosedimentary data and I think this manuscript is acceptable for publication with minor revision.

My main suggestions are intended to provide additional information to help those who would like to use these data sets or replicate similar collection and processing procedures. In addition to minor editorial suggestions, they focus on the following two points: (i) the description of the data processing and qualification procedures, which is sometimes too brief or a bit confusing, and (ii) how the quality codes assigned to the different data can or should be used.

Answer: we have added information in the revised version concerning the qualification procedures, and how to estimate quantitatively the uncertainty associated with the data to use these quality codes.

My suggestions for improving the paper are detailed below in three comment sections (general/specific/editorial)

**General Comments:**

All the data proposed in this document have been qualified with a quality code -presented in Table 2in the form of 4 levels (good - intermediate - uncertain - poor) if we exclude the codes for missing data and data without quality code. The effort made by the data producers to generate such detailed quality grades and to attempt to formalize the attribution rules is noteworthy. However, nothing is said about how to take these different levels into account when analyzing these data. It would be greatly appreciated if the authors could provide guidelines to future users of the data on how to take these different levels into account. For example, do they advise to associate different levels of uncertainty to them? If yes, could the authors provide an idea of uncertainties associated to each level (at least as a range, i.e. [Min; Max]) ? Since uncertainties may depend on the configuration of each data monitoring system, it would probably be appropriate to include this information in the header of each data file. In addition, I would suggest including in paragraph 4 a small discussion of how existing studies that have used these data have accounted for these different levels of quality...

Answer: We thank the reviewer for these suggestions.

We have added information in the revised paper on the meaning of quality codes and details about how to estimate quantitatively the uncertainty that should be associated with these codes in each

data section (lines 117-116, 157-159, 273-274, 343-367, 425-429 in the tracked change revised version). We hope this information will be useful for the reader to re-use the data.

Concerning data download, every download from BDOH is already accompanied by a description of the quality codes, and we have added the detailed information on what this code means for each data set in BDOH itself.

Concerning previous studies, we thank the reviewer for this suggestion. We were not able to collect detailed information on how quality was used from most of the previous studies that are listed in the paper. In general, previous studies tended to exclude events, or years, with poor quality or missing data, in their analysis. We have mentioned it line 114: "Previous quantitative analysis on Draix observatory dataset generally excluded such data"

**Specific Comments:**

L41-42: I would not use "are only available in..." as other datasets combining records of suspended and bedload may exist (e.g., DOI: 10.2478/johh-2019-0003).

**Answer: we have removed "only" from the sentence as suggested**

L85-86. The sentence "Mean annual rainfall for the observatory... and mean annual temperature..." would require you to provide the method you used to estimate a average value from multiple measurement locations. Instead, I suggest that you provide an average value at a single observatory location inside the observatory and indicate the time period used to estimate the average.

**Answer: We agree with the reviewer suggestion and we have indicated the mean annual rainfall at the Laval raingauge over the period 1985-2022 and mean temperature at the Plateau station over the period 2001-2022**

L96-97, Table 1 and Legend of Figure 1: At the end of the introduction the authors explain that the data from the Galabre catchment area, which belongs to the Draix-Bléone observatory, will not be presented in this datapaper because they have already been published. Therefore, I suggest not to mention it again here, nor to present it in Table 1, nor to mention that it is not represented in Figure 1.

**Answer: We agree with the reviewer and we have modified the manuscript and table 1 as suggested**

Table 1: please consider i) dropping the line for Galabre; ii) adding a column indicating the % of badland area for each catchment; iii) providing one sentence to describe how the average catchment slope was evaluated (from which data ? which spatial resolution if derived from DEM...); iv) specify in Line 91 that the vegetation cover was stable during the whole period of monitoring (if not, provide a range of value instead of a unique value)

Answer: we have removed the line for Galabre. The % of badland area is 100% for all catchments except for the Bouinenc, for which we do not know the % of badland area. This is indicated in the texte as follows: "All catchments drain only marly badland areas, except for the larger catchment, the Bouinenc, that integrates a wider variability of lithologies and land cover"

We have recomputed the average catchment slopes using a recent DEM and indicated the origin of catchment slopes in the table caption: "Average slope derived from 5-m DEM from IGN BDALTI 2016"

**Table 3: add the resolution for the Archail raingauge**

**Answer: added as suggested**

L129-131 & 140: please provide more details on how the raingauge data were corrected because I didn't see in the data I consulted any adjustments of the bucket volume.

Answer: When there is a misfit between the bucket volume and the recorded rainfall, and if the misfit is less than 10%, the data from the raingauge is replaced by the corrected data. Therefore the data which is available in the data base is the corrected data only. For a misfit less than 10%, the corrected data is considered of good quality and is given a quality code 2. We have added the following in the revised paper: "When there is a misfit smaller than 10% between the recorded rainfall and the container, the data that is stored in the database is the corrected data with a quality code "2"."

Figure 5: please consider using a ratio 1:2 between y1 and y2 (i.e. P = 2\*T) as usually done to present this king of graph in the Mediterranean context (here you used P = 4T)

**Answer: this has been changed as suggested in the revised version**

Table 5: add the acquisition frequency for the Bouinenc station

**Answer: Added as suggested**

L229 & 233: Please specify the difference between noise suppression (L233) and oscillation suppression (L229) and/or consider merging the two steps.

**Answer: we have merged the two steps as suggested**

L234-235: Please provide more detail on how the time series of recorded water levels are corrected for inconsistencies between some occasional scale readings (or manual bucket measurements) and the continuously recorded data.

Answer: we have added the following in the revised text (line 239): "If there are several scale readings or bucket measurements that indicate a similar misfit with the sensor data, the sensor data is corrected to fit the manual record (shifted), otherwise, the sensor data is kept with a lower quality code."

Section 3.3.1: Can you provide a table summarizing the data set for suspended load, as was done for all other data categories?

Answer: we have added this table in the revised paper (now Table 7).

L285-286: How many samples are enough to establish an event-specific relationship?

Answer: we have indicated in the revised version that we build such relation when at least 4 samples are available for one flood

L285-295: the combination of procedures (and the priorities between each step) need to be clarified (see comments below).

Answer: we agree that this workflow needs to be explained better, so we have rephrased the text in the revised version and added a figure to make it clearer (now Figure 8)

Step 2: Consider merging step2 into step4.

Answer: We think that the steps should be kept separated. We have added a figure to explain the procedure and hope it makes it clearer

Step 3: is is not clear if all sediment concentrations derived from the voltage/concentration (or the turbidity/concentration) relation are included in the reconstructed concentration time serie. If only part of them are included, please detail the criteria used. What happens if no sample is available ?

Answer: We do not use all the concentrations derived from the turbidity/concentration relation, rather we only use the points that are needed to describe the flood dynamics where no samples are available (flood peaks, inflexion points in the hydrograph, long periods without samples). This procedure was designed to give more credit to sample data rather than turbidity data, such that estimated concentration is always equal to sample concentration where samples are available.

If no sample is available, then we simply use the annual or interannual turbidity/concentration relation. We have added the following in the revised paper: "On the other hand, when no samples are available for a flood, we simply use the annual calibration curve to convert turbidity into SSC".

Step 4: Are concentrations derived from discharge/concentration relations included only when turbidimetric measurements are not available (i.e., shallow water depth, or concentration below 10 g/L, or device failure) ? are there other cases where they are included in the final reconstructed concentration dataset ?

Answer: we first build the discharge/concentration diagram using all samples available and a few points from the turbidimeter. Then we interpolate between these points, which is equivalent to building piecewise linear relations between discharge and concentration. This was found to be the best option to combine turbidity and sample data, ensuring that estimated concentration is equal to sample concentration where samples are available, while conserving the hysteretic pattern between C and Q. We have added a figure to explain this better in the revised paper.

Step 5: What is the decision criterion for moving from the Step 4 to the Step 5?

Answer: This usually happens at the end of the flood when we do not have anymore concentration data and need to extrapolate the decrease in concentration. This is not a very accurate estimation but it represents a small portion (less than 1%) of the total flood sediment export.

L300: consider changing "volume" by "mass" in this sentence, as multiplying the discharge (I/s) and the concentraztion (g/l) provides a flux (g/s), and then integrating gives a mass.

**Answer: done as suggested**

L301: consider switching "volume" and "mass" in this sentence.

**Answer: done as suggested**

L304: "When enough samples are available" is vague... please try to provide an idea on how many samples are required ?

Answer: we have added more information concerning the procedure and quality assessment, and in particular have included the following information: "When at least 4 samples are available and are well distributed over the flood duration and over the range of discharges, an event-scale calibration is used and combined with the sample data and the resulting SSC time series is given quality code 2"

L313-314 & legend of Figure 7: please specify whether instantaneous or event-scale average concentrations are concerned

Answer: we have added "instantaneous" in the revised text and legend of figure 7

L352-354: please explicit the rule(s) for assigning a volume of sediment deposits to a series of floods.

Answer: the volume is attributed to the series of floods that occur between two consecutive topographic surveys of the trap, as written in the text: "the volume is attributed to the batch of floods that occurred between the two consecutive measurements of topography"

L364: Would it be possible to mention the long-term bedload's contribution to total exports (in addition of the value for 2013) ?

**Answer: we have added the following: "Averaging over all the period 2003-2022, the respective contributions of suspended load and bedload for the Laval station are 32 and 68%"**

Section 4: as already said, the author have extensively illustrated, via previously published studies, the potential of the proposed dataset to contribute to a variety of issues related to runoff and erosion in this specific mountainous environment. Surprisingly, the authors did not provide perspectives on potential future uses of their datasets. Please could you discuss a bit more potential future works. I also suggest to ad one or two sentences on the future of data collection.

Answer: concerning the future uses of the dataset, we had already indicated that "Predicting sediment yield on such catchments, and specifically at the event scale, is therefore still an open question, which requires both a better understanding of slope processes and appropriate tools to simulate the dynamics of flood events" and that "The long record of water and sediment fluxes for these two contrasted catchments is clearly an opportunity for a better understanding of the effect of vegetation on the water cycle and on erosion processes" and mentioned the "potentiality of the dataset presented here to contribute to a variety of questions concerning critical zone processes and the interactions between chemical, physical and biotic components in these badlands".

We have added the following in the revised paper concerning future studies: "Future studies may therefore focus on exploring modelling strategies to improve sediment yield and erosion prediction, from the event-scale to longer time-scales, tracking the effects of climate change on hydrology and sediment processes in the last 40 years in order to better predict future badland response, addressing the coupling between vegetation and erosion, subsurface water flow, physical and geochemical weathering, etc."

And we have added the following concerning the future of data collection: "Data collection will continue since the observatory has been labelled as SNO (Service National d'Observation) meaning that French institutions (CNRS and INRAE) are willing to fund it for long-term observation. Objectives for the next years include maintaining existing datasets and particularly securing bedload measurement at the Laval station, validating and making available existing datasets (e.g. soil moisture and rain drop size and velocity distributions), developing water chemistry measurements

on a regular basis, testing surface image velocimetry to refine the estimation of the highest discharges, and build detailed maps of vegetation cover on the catchments".

Section 5: I haven't found a way to access GIS files, such as catchment boundaries or device locations, either in BDOH or in the 5 year snapshot provided for the review process. If these data already exist somewhere in access, I suggest mentioning the links to access them, otherwise I suggest adding them to the datasets proposed in this datapaper.

Answer: We thank the reviewer for this very useful suggestion and we have added a link in the paper asset to the spatial data for the observatory including DEM information, instrument location and catchment boundaries

**Editorial Comments:**

L64: Drop "^" from "Legoût"

Answer: done

L333: Change "Fig.9" into "Fig.8"

**Answer: done**

References: please carefully check the list of references as I identified the following issues: i) (Le Bouteiller et al., 2019) and (Gras et al., 2007) to be added ; ii) alphabetical sorting problem for Klotz at al.; iii) reference (Olivier et al., 1995) is presented in two parts (L534 & 553-554)

Answer: this has been corrected in the revised paper

**Response to reviewer 2**

This is a nice paper giving a comprehensive overview on the infrastructure, data and measurements of the Draix-Bléone observatory. I have two main comments on potential additions, which the authors may want to think about.

First, I think it would be helpful to represent some of the information graphically, for example, some of the more complicated workflows (suspended load), and the history of infrastructure and instrumentation. The latter could be done as a Gantt chart, showing timelines of deployment for different stations and their sensors, and highlighting changes in instruments and procedures.

Answer : a Gantt chart has been added with the acquisition periods for each variable and station (now figure 3)

Second, the topic of errors and uncertainties is hardly treated at the moment. The authors explain the quality assessment into categories; yet this is something different to giving quantitative uncertainties. How do the different classes convert to errors that could be added to plots? Have some of the uncertainties been systematically assessed, for example through repeat measurements or by comparing to high-quality benchmarks? For example, the suspended load measurements using the turbidity probes can be compared to the benchmark data from manual sampling to provide error estimates.

Some more comments can be found below.

Answer: We have added information concerning the estimation of the uncertainty for each type of data in the revised paper. In particular, we have explained better the meaning of each quality code in the introduction of section 3, then we have detailed how to estimate an uncertainty for rainfall, meteo data, discharge, suspended sediment and bedload volumes (lines 117-116, 157-159, 273-274, 343-367, 425-429 in the tracked change revised version).

Concerning the suspended load measurements, we have indeed used the comparison between samples and turbidity measurements to quantify the uncertainty of SSC data obtained from a turbidity calibration relation. This uncertainty is high for the lowest concentrations since our sensors have been designed for the extremely high concentrations that can be encountered on this site. However, we underline that the methodology we use to reconstruct SSC time series gives more credit to samples than turbidity data, and that the effort devoted to collect and process a large number of sample allows us to reduce the uncertainty compared to a traditional method based on the turbidity/concentration.

21 here and elsewhere: dot missing after et al. (et al. is an abbreviation for the latin 'et alia', which means 'and others')

Answer: corrected in the revised paper

33 an 'and' missing before the final item of the list.

Answer: added as suggested

39 Could refer to Turowski et al. Sedimentology 2010 here.

Answer: thank you for the suggestion, reference has been added

51 I agree with the statement here. Yet, the cited literature does not give justice to the important scientific results coming out of badland studies. There are key papers for example of Alan Howard, or from Chinese and Taiwanese research sites that might be added.

Answer: we have added references to Howard and Kerby 1983 and Higuchi et al. 2013

61 ... consists of ...

Answer: changed as indicated

178 ... while others...

Answer: changed as indicated

186 maybe show the different sensors graphically, in a timeline or as a Gantt chart?

Answer: we thank the reviewer for this suggestion and we have indeed provided a Gantt chart to synthetize the available data. However we have chosen not to indicate the type of sensors in this chart to keep it simpler since there has been quite a few changes in the long history of the site, and also because we consider that the uncertainty related to changes in instruments is negligible compared to other sources of uncertainty.

191 What is the threshold?

Answer: we have indicated in the revised version that the threshold was 5mm

197 ...during each field visit.

**Answer: changed as suggested**

198 Again, it might be helpful to have a graphical representation of the activities / data available for the various gauging stations.

**Answer: as said earlier we have added a Gantt chart in the revised version**

200 please add references for the Parshall standard relation and the depth-discharge relation of the trapezoidal flume or provide more information.

Answer: we have added a table with the equations for the depth-discharge relation at each station (now table 6)

**205 How is discharge obtained for the Roubine?**

Answer: discharge is obtained using a weir relation now listed in table 6. We have added the following in the revised text: "The discharge is computed from the water depth using the weir equation listed in Table 6"

215 ... based on manual measurements of flow velocity using the salt dilution method.

Answer: changed as suggested

**215 please give the date of the flood**

Answer: we have indicated in the revised paper that this happened in June 2013, and that it was due to a series of floods rather than at one particular date.

217 change to "As a result, no discharge data is available after this date" or similar.

Answer: changed as suggested

258 Can you give details on the thresholds?

Answer: we have included some information about the thresholds for the sampling, however these are specific to each station, therefore we have only indicated as an example the values of the thresholds for the Laval station. Text added: "when water depth exceed a threshold (e.g. 20 cm at Laval), remains higher than this threshold for more than half an hour, and when water depth changes faster than a threshold (e.g. 10 cm per minute at Laval)"

262 turbidity

Answer: changed as suggested

271 suction pipe

Answer: changed as suggested

273 a graphical representation of the work flow may be helpful

Answer: we have added a figure to better show how we proceed to reconstruct the concentration time series

286 ... is used.

**Answer: changed as suggested**

314 An implication from the statement here is that, at high concentrations, viscosity and density of the fluid significantly change. Is this considered when converting stage to discharge? If yes, how? If no, what is the potential error?

Answer: yes the reviewer is right that this effect could impact the depth-discharge relation. We have not modified it however since the potential error appears to be limited to the first instants of the flood (when depth is low and concentration is high) according to our tests. We have added the following explanation in the revised paper: "This effect has not been accounted for in the depthdischarge relations and might affect the discharge estimation in the earlier moments of highly concentrated floods. However, we tested this effect for the flood of July 6th, 2006 at the Laval station, and found that the resulting error at the flood-scale was less than 1% for the total water volume and less than 3% for the total sediment volume"

330 maybe add a brief discussion on how the traps affect discharge measurements and the relationship between discharge and bedload transport (they probably act as a buffer for high flood peaks).

Answer: we have mentioned this possibility in the revised paper, in the section concerning the quality of the discharge data: "Another source of possible uncertainty for the highest flood peaks could be a buffer effect due to the sediment trap located upstream of the station". However we do not have much information to discuss it more in detail here.

**342 How is the total volume measured?**

Answer: this total volume excavated is measured as a number of buckets for the Roubine (which is manually excavated) and as a number of trucks for the Laval and Moulin (excavated with a power shovel).

We have added the following in the revised article: "number of buckets or number of trucks filled with sediments"

344 Can you give typical values?

Answer: we have indicated in the revised paper that this was less than 20%

**Response to reviewer 3**

"A high-frequency, long-term data set of hydrology and sediment yield: The alpine badland catchments of Draix-Bléone Observatory" proposed by Klotz et al. is a paper that presents a quasiunique dataset consisting of long-term measurements of rainfall, meteorology, runoff, suspendedand bed-load in the Draix-Bléone Observatory. The materials and methods used to realize the monitoring program were described in detail, also making available a 5-year dataset via the BDOH web interface. As stated by the authors, long-term datasets about climate, hydrological, and sediment dynamics are rare, especially in the alpine range. Therefore, efforts such as this to present in detail the processes of data acquisition and elaboration are very valuable, especially if the data are then made available to the scientific community, as in this case. That said, I have some comments. First, it is unclear (to me) what message the authors want to transmit with this article. I am not sure that the presentation of a dataset and the relative acquisition procedures are sufficient for publication. Maybe, an interesting aspect would be how the uncertainties and the quality code presented may affect the results obtained. This point could be discussed in section 4, which sounds highly site-specific.

Answer : The message that we want to deliver is that we make this dataset available to the community since we think it is of sufficient originality and quality to be reused and improve our collective understanding of hydrological and sediment transport processes. We think this fits into ESSD scope (i.e. "Earth System Science Data (ESSD) is an international, interdisciplinary journal for the publication of articles on original research data (sets), furthering the reuse of high-quality data of benefit to Earth system sciences. The editors encourage submissions on original data or data collections which are of sufficient quality and have the potential to contribute to these aims").

To ensure that the dataset can be reused and particularly for modeling purpose, we will make available, as suggested by another reviewer, the geographic information of the catchments (DEM and catchment contour shape files)

Moreover, following the reviewer's comment on uncertainty, we have also added more detail in the revised paper concerning the quality codes and how they have been, and should be, taken into account when using the data (lines 117-116, 157-159, 273-274, 343-367, 425-429 in the tracked change revised version).

My specific observations, comments, and suggestions are:

Table 2: I found the code a bit misleading. Maybe, it would be better to assign 0 to missing data, 1 to no quality, 2 to poor quality.....5 to good quality. So, it might be easier to follow?

Answer: This quality code was defined a long-time ago and has been implemented as such in the BDOH database, therefore it is not possible to change it now (or it would require a large modification of the database for which we do not have resources now). Note however that every data download from BDOH database is accompanied by a readme file that explains clearly this data quality code for the user.

L129-130: Was this correction made by spreading the difference over the entire period or by considering only the rainfall events? Please, specify.

Answer: The difference was spread only on the antecedent rainfall period. This was modified in the revised paper: "and the data for the antecedent rainfall period is corrected such that the cumulated rain over the period corresponds to the real water volume that was collected"

Table 4: Mean wind speed and Mean wind direction are mean values as Temperature, Humidity, and Radiation? If yes, please add such information in the table as made for the other variables.

Answer: yes, mean wind speed and mean wind direction are mean values and this has been added in Table 4 as suggested by the reviewer.

L186-189: Could the use of different devices have caused a certain uncertainty in the data gathered? If yes, please discuss this point.

Answer: No we don't think that the use of different devices has caused significant uncertainty in the data, or it would be negligible compared to other sources of uncertainty. For instance, for water depth, any type of instrument is now able to measure water depth with an accuracy smaller than 1cm, but surface ripples are generally of this size, therefore the uncertainty due to ripples is higher than the uncertainty related to the instrument

L194: So, you visited the study areas every 2-4 days?! Impressive! Maybe, it could be interesting to spend some words about the pros and cons of this high sampling resolution.

Answer: The observatory has a technician on site since 2000, who visits the stations every 2-4 days. This is obviously a great advantage in terms of data quality, and it makes it possible to repair or replace quickly the monitoring devices, reducing the periods of « missing data ». There are not really drawbacks apart from the cost for the research institute in terms of human resources. We have added the following in the revised paper : « Regular field visits are a great advantage for data quality, since they allow manual measurements of low discharges and scale reading, and observations, that are useful to interpret, criticize and possibly correct the sensor data"

L199-200: "Note that...measurements" could be removed. Unnecessary.

Answer: this has been removed in the revised version

L227: Nilomètre should be Nilometer? Please, check over the entire manuscript.

Answer: We were not aware that the word "nilometer" existed, that's why we initially kept the French name "nilomètre". We have changed it into "nilometer in the revised version

L313-314: Please, rephrase. Unclear.

Answer: we have rephrase this as follows: "The highest concentrations are observed at the Laval station, and instantaneous concentrations higher than 600 g L-1 are have been observed 11 times for the last 30-years period "

L435: Was this emphasis given in the dataset or in the article? In both cases, the sentence sounds quite cryptic. Please, rephrase.

Answer : we detailed earlier in the paper that the suspended sediment concentration was only measured during floods, that represent more than 99% of the annual sediment flux. That is the reason why we mention here that the dataset has a particular emphasis on floods. We have rephrased as follows: "data spans up to 40 years, with a particular effort devoted to measure floods, since these short periods are responsible for most of the sediment fluxes".